# Can Custom Models Learn In-Context? An Exploration of Hybrid Architecture Performance on In-Context Learning Tasks

## Abstract

In-Context Learning (ICL) is a phenomenon where task learning occurs through a prompt sequence without the necessity of parameter updates. ICL in Multi-Headed Attention (MHA) with absolute positional embedding has been the focus of more study than other sequence model varieties. We examine implications of architectural differences between GPT-2 and LLaMa as well as Llama and Mamba. We extend work done by Garg et al. (2022) and Park et al. (2024) to GPT-2/LLaMa hybrid and LLaMa/Mamba hybrid models – examining the interplay between sequence transformation blocks and regressive performance in-context. We note that certain architectural changes cause degraded training efficiency/ICL accuracy by converging to suboptimal predictors or converging slower. We also find certain hybrids showing optimistic performance improvements, informing potential future ICL-focused architecture modifications. Additionally, we propose the "ICL regression score", a scalar metric describing a model's whole performance on a specific task. Compute limitations impose restrictions on our architecture-space, training duration, number of training runs, function class complexity, and benchmark complexity. To foster reproducible and extensible research, we provide a typed, modular, and extensible Python package on which we run all experiments. This code is available at `https://github.com/anonymousforneurips64/neurips2024-submission21757`.

## 1   Introduction

Popularized by Large Language Models such as GPT-2 [1] and GPT-3 [2], In-Context Learning (ICL) is the ability for highly expressive generative sequence models to predict phenomena by processing demonstrations without performing traditional gradient steps. Such phenomena vary from effective control systems [3] to answering questions in natural language [4, 5]. A large body of recent work has studied this phenomenon in transformer models [6, 7, 2, 1, 8, 9, 10, 11, 12, 13, 14, 15, 16, 17, 18, 19, 20, 21, 22, 23, 24, 25] , which derive in structure from Vaswani et al. [26].

Some recent examples of this research on ICL include Garg et al [6], which studies ICL by providing a variety of function classes for models to learn, additionally benchmarking robustness by testing performance on out-of-distribution data. Guo et al[11] shows the validity of composing simple function classes to produce complex ones, while Liu et al [20] produced a metric for model information recall. These works give us a set of metrics with which we can use to compare model performance on ICL.

ICL was initially primarily studied in attention-based models but has recently been explored in other sequence models, creating discussion on its differences across those models and why these

| Task | dim ($d$) | points ($N$) | $x$ distribution | $y$ calculation / parameter distribution | Task-specific |
|---|---|---|---|---|---|
| Linear Regression | 20 | 41 | $\mathcal{N}(0, I_d)$ | $w \sim \mathcal{N}(0, I_d)$ | – |
| Sparse Linear | 20 | 41 | $\mathcal{N}(0, I_d)$ | $w \sim \mathcal{N}(0, I_d)$, $\texttt{sparsity}(w) \leftarrow k$ | $k = 3$ |
| 2-Layer MLP | 20 | 101 | $\mathcal{N}(0, I_d)$ | $W_{ij}^{(1)}, W_{ij}^{(2)} \sim \mathcal{N}(0, 1)$ | width $= 100$ |
| Decision Tree | 20 | 101 | $\mathcal{N}(0, I_d)$ | $\text{leaf} \sim \mathcal{N}(0, 1), \text{non\_leaf} \sim \{1, ..., d\}$ | depth $= 4$ |
| Sparse Parity | 10 | 140 | $\{-1, 1\}^d$ | $y = \prod_{j \in I} x[j]$ | $k = 2$ |
| Vector MQAR | 20 | 128 | $\text{Unif}(\mathcal{S}^{d-1})$ | $y \sim \text{Unif}(\mathcal{S}^{d-1})$ | – |

Table 1: Summary of tasks. Each regression target $f_\theta(x_i)$ is either parametrized by a randomly sampled $\theta$ or directly computed/sampled as detailed above.

occur architecturally. In our paper, we study this by substituting key modern transformer (Llama) components with Mamba blocks and GPT-2 components and richly benchmarking.

Since ICL for complete natural language understanding often requires training models with over a billion parameters, the effects of architectural changes on fine-grained ICL abilities are often left unexplored. As a consequence, although language models have progressed quickly and entertained radically new architectures, there is limited extensible research that explores the effects of fine-grained architecture choices on ICL ability [8, 14]. Garg et al. established using simple function classes to evaluate ICL ability and examined solely GPT-2 as a sequence model. Lee et al. [8] expanded this analysis on a slightly different set of function classes for a variety of base models. Park et al. [14] evaluated ICL performance of 2 hybrid architectures between Mamba and GPT-2. Using unmodified Llama/Mamba/GPT-2 as a control, we analyze GPT2-Llama and Llama-Mamba hybrid architectures derived from replacing portions of GPT2 components with analogous Llama sections and LLama with Mamba blocks, respectively, in 12 total architectures (3 unmodified + 9 hybrid).

We observe that the code written to analyze ICL with simple function classes – although almost unanimously extensions of Garg et al.'s – often requires substantial, structural changes to the parent codebase[1], greatly heightening the barrier to extending each project in turn. Inspired by Donoho's ideal of Frictionless Reproducibility [27], we provide a set of simple abstractions and interfaces to facilitate extensions and modifications to our code while promoting interoperability between forks.

## 2 Related Work

There are many ways to capture qualitative aspects of ICL with quantitative measures. Weber et al. [17] compare the agreement between generations of a language model under varying prompts of equal meaning to test robustness to variations. Olsson et al. [22] compute a heuristic "ICL score" to measure an accuracy increase in predictions of a model given more context. We adapt this metric to fit our experimental setup more aptly, regularizing along both the number of in-context examples and against a baseline predictor.

In general, evaluating ICL ability has been approached from two primary avenues: both when the only solution at train time is to meta-learn an algorithm [6, 8, 28, 11, 19] and when optimal loss at train time can also be satisfied by memorization or otherwise leveraging previously trained-on data [10, 23]. In this work, we take the former approach through learning a regression algorithm to randomized simple function classes [6, 11, 15].

Further still, non-transformer architectures are capable of ICL [8]. Lee et al. [8] observed ICL in numerous sequence model architectures (e.g. RNNs, Mamba, S4, CNNs, GPT-2, and Llama) and found qualitative differences in each architecture's performance. Chan et al. [25] found that Transformers depend on "burstiness" and long-tail distributions of natural data to outperform RNNs and LSTMs in ICL tasks. Park et al. [14] uses simple function classes similar to Garg et al. [6] in evaluating the ICL ability of Mamba, S4, S4-Mamba, and GPT-2. They find an overlapping but inequivalent set of function classes for which each model succeeds and construct a hybrid architecture

---

[1]As mentioned, our code takes notable inspiration from the code distributed by Garg et al. [6], Park et al. [14], and Lee et al. [8], which can be found at `https://github.com/dtsip/in-context-learning`, `https://github.com/krafton-ai/mambaformer-icl`, and `https://github.com/ivnle/synth-icl` respectively. The first two repositories are licensed under the MIT License and we could not identify the license for the third.

to achieve the union of these abilities. We further this work by closely examining the contributions of individual architectural changes for GPT-2 and Llama-style transformers towards ICL ability.

## 3   Methods

As established by Garg et al. and extended by recent work, our ICL tasks take the following form [6, 8, 14]:

$$\underbrace{x_0, f_\theta(x_0), x_1, f_\theta(x_1), ..., \overbrace{x_N}^{\text{query}}}_{\text{prompt } P}, \underbrace{f_\theta(x_N)}_{\text{completion}}$$

where $P$ is a series of input-output pairs followed by a lone query. The model predicts a completion based on the prompt it received. The function parameters $\theta$ and the inputs $x_i$ are randomly sampled from a function class domain and an input domain, respectively. The tasks we regress to are summarized in Table 1 and detailed in Section 3.1

We train models for ICL by minimizing the expected loss over a distribution of prompts and corresponding function outputs. This approach allows us to observe qualitative differences in model architectures by their ability to behave similarly to optimal or baseline estimators. To further simplify ICL aptitude evaluation, we introduce a proxy value summarizing a given model's ICL ability for a specific task. This metric averages the error of a model normalized by the baseline error at each context length. We detail this further in Section 3.3.

### 3.1   Training

To determine task-specific ICL ability, our sequence models regress onto the functions shown above [14]. We replicate the function classes `Linear Regression`, `Sparse Linear Regression`, `2-Layer MLP Regression`, and `Decision Tree Regression` from Garg et al. [6] as they present a wide range of "difficulty" for sequence models. In addition, to capture the existence of some ICL ability, we also regress onto the two function classes examined in Park et al. [14]: parity function with induced sparsity (`Sparse Parity`) and parallel associative recall (`Vector MQAR`).

Unless otherwise specified, we train all models with 12 layers, 8 attention heads, an expansion factor of 4 (in the case of models with Mamba Mixer layers), and linear layers to transform the input sequences into and from the embedding dimension of 256. We use the ADAM optimizer with a learning rate of 0.0001 for 500k steps. Our expansion factor was selected to ensure similar parameter counts across baselines and all other hyperparameters were chosen for consistency with Garg et al. [6]. Note for the four function classes from Garg et al., the same curriculum was used during training. No curriculum is used for the two new function classes from Park et al. [14]. For our compute[2], we utilized 898.90 hours on an A10, 55.74 hours on an RTX 3090, 151.90 hours on an RTX 4090, 75.48 hours on an RTX 4070 Ti, and 9.83 hours on an RTX 6000.

**Linear Regression and Sparse Linear Regression** Each function in these tasks is parametrized as a single weight vector ($w$) of dimension equal to that of the $x$-values (i.e. 20) so that $y = w^T x$. We sample the coordinate values from a normal distribution and (in the Sparse Linear case) zero out all values except a uniformly at random selected $k$ coordinates. In essence, one can consider Linear Regression to be the degenerate case where the $k = 20$. We preserve these tasks from Garg et al. [6] to verify that none of our hybrid modifications lose the near-optimal performance that was already found with GPT-2.

**2-Layer MLP Regression** We fill two weight matrices $W^{(1)} \in \mathrm{R}^{100 \times 20}$ and $W^{(2)} \in \mathrm{R}^{1 \times 100}$ with scalar samples from a normal distribution. $y$ values are computed as the result of a forward pass through a 2-layer multi layer perceptron with a ReLU activation. That is: $y = W^{(2)}\texttt{ReLU}(W^{(1)}x)$. This is a more complex function class that Garg et al. [6] found that GPT-2 can perform very well at, suggesting that this task can capture some ICL ability of an architecture.

---

[2]On an A10, the approximate training time for `Linear Regression` and `Sparse Linear Regression` was 12 hours, for `2-Layer MLP Regression` and `Decision Tree Regression` was 2 days, and for `Vector MQAR` was 5 hours.

114 **Decision Tree Regression** We construct full decision trees of depth 4 with leaf values sampled from a
115 normal distribution and branching conditions to be selected uniformly at random over the coordinates
116 of the input dimension. The left branch is taken if the selected input coordinate is less than 0 and the
117 right branch is taken otherwise. Garg et al. [6] found that GPT-2 was able to achieve much lower
118 error for lower context lengths than XGBoost or Greedy Tree Learning, suggesting that this task can
119 capture some ICL ability of an architecture.

120 **Sparse Parity** We select $k = 2$ values to consider and compute their parity, expressed as either $-1$ or
121 1. That is, we uniformly sample without replacement $\theta \sim \{1, ..., 10\}^k$ and compute $y = \prod_{i \in \theta} x[i]$.
122 Along with a higher learning rate of $0.0004$, this is identical to the scheme implemented in Park et al.
123 [14]. They [14] found that GPT-2 style transformers do not perform well on this task, suggesting that
124 this is a discerning proxy for measuring ICL ability. Finally, as convergence was quick for this task,
125 we only trained models up to 200k steps.

126 **Vector MQAR** We sample $2N$ points from the $d$-sphere of radius $\sqrt{d}$ and group them randomly into
127 pairs to forming $N$ key-value pairs. For consistency with the experiments of Park et al. [14] and to
128 reliably allow for the formation of transformer circuits highly relevant to this task [22, 14], we reduce
129 model complexity by using an embedding dimension of 128, 2 layers, and a higher learning rate of
130 0.0002. Park et al. [14] found that Mamba, our representative of SSM-type models, performed poorly,
131 suggesting that this task can serve to ensure we don't lose capabilities provided by transformers.

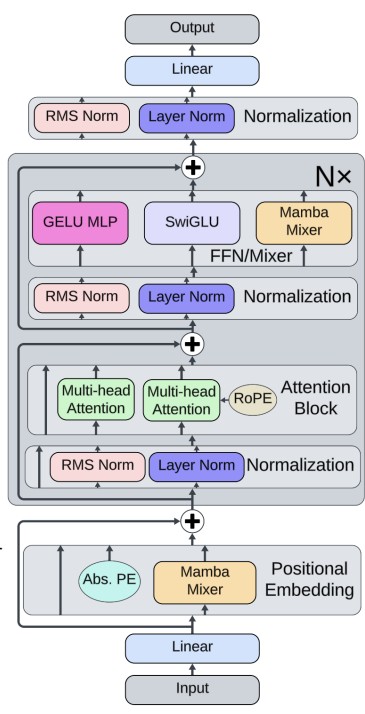

| | Model Variation | Pos. Emb. | FFN | Normalization |
|---|---|---|---|---|
| (1) | GPT-2 | Absolute | GELU MLP | Layer Norm |
| (1.1) | GPT-2 RMS | Absolute | GELU MLP | RMS Norm |
| (1.2) | GPT-2 RoPE | RoPE | GELU MLP | Layer Norm |
| (1.3) | GPT-2 SwiGLU | Absolute | SwiGLU | Layer Norm |
| (1.4) | GPT-2 RMS SwiGLU | Absolute | SwiGLU | RMS Norm |
| (1.5) | GPT-2 RMS RoPE | RoPE | GELU MLP | RMS Norm |
| (1.6) | GPT-2 RoPE SwiGLU | RoPE | SwiGLU | Layer Norm |
| (2) | Llama | RoPE | SwiGLU | RMS Norm |
| (2.1) | Llama RoPE-less | Mamba Mixer | SwiGLU | RMS Norm |
| (2.2) | Llama SwiGLU-less | RoPE | Mamba Mixer | RMS Norm |
| (2.3) | Llama RoPE,SwiGLU-less | Mamba Mixer | Mamba Mixer | RMS Norm |
| (3) | Mamba | – | Mamba Mixer | RMS Norm |

(a) For our hybrid architectures, we modify 3 types of architectural sub-blocks: positional embeddings, feed-forward network, and normalizations. We specify the sub-block alternatives used for each architecture.

(b) A block diagram illustrating how each variation affects the overall architecture. Note that vertical arrows in a given block indicate that some variations skip that block entirely.

Figure 1: Visual aid for our explored hybrid models in tabular and graphical format.

## 3.2 Architectures

133 As detailed by Radford et al. [1], GPT-2 is almost identical to the original decoder-only transformer,
134 with absolute positional embedding, pre-norm layer normalization, and a GELU activation function
135 in the feed-forward network (FFN) (which is otherwise a multi-layer perceptron). In contrast, Llama
136 [29, 30] combines a number of modern transformer modifications, including swapping layer norm
137 with RMS norm [31], changing the architecture and activation function of the FFN, and using rotary

| | GPT-2 | Llama | Mamba |
|---|---|---|---|
| Positional Embedding | Absolute | RoPE | None |
| Feed Forward Network | 2 layer MLP | Convolutional MLP | None |
| Attention Mechanism | Multi-Query Multi-Head | Multi-Query Multi-Head | Mamba Mixer |
| Normalization | Layer Norm | RMS Norm | RMS Norm |

Table 2: A summary of the primary architectural differences between GPT-2, Llama, and Mamba. We examine all variations between GPT-2 and Llama and all variations between Llama and Mamba.

positional embeddings instead of absolute positional embeddings [32]. We acknowledge that the larger variations of Llama2 [30] and both variations of Llama3 [33] used Grouped-Query Attention (GQA), however we surmise that at our model scales of $\sim$10 million parameters, GQA will not significantly affect the performance of our models. From an entirely different method of sequence modeling, Mamba forgoes positional embedding entirely, combining features of the Gated Linear Unit and state space expansion to remove the need for distinct attention and feed-forward blocks. We summarize these architectural differences in Table 2. We examine all combinations of these different components, training 12 total architectures (listed in Figure 1a) on our 6 tasks for a total of 72 model-task pairs. Figure 1b illustrates how each of these variations compose into a model. We provide individual diagrams of each architecture in Appendix A.

## 3.3 Evaluation

In addition to the baseline metric (squared error as a function of context length) from Garg et. al. [6], we've established another metric: ICL regression score. This is a scalar expressing overall performance of a model on a task. Abstractly, the metric aims to capture the proportion of the baseline error saved by a model. The regression score is calculated by (1) computing the difference in error achieved by the model and the zero estimator at each context length, (2) computing the average of this value over the length of the sequence, (3) computing the same value for the baseline estimator, and (4) taking the ratio of these.

In summary, ICL regression score can be calculated as follows:

$$S_{\mathrm{model}} = \frac{\sum_i \left( \xi_{\mathrm{model}}^{(i)} - \xi_0^{(i)} \right)}{\sum_i \left( \xi_{\mathrm{base}}^{(i)} - \xi_0^{(i)} \right)} \tag{1}$$

where $\xi_{\mathrm{model}}^{(i)}$ is the squared error of the model of interest at context length $i$. Sim. $\xi_{\mathrm{base}}^{(i)}$ for baseline and $\xi_0^{(i)}$ for the zero estimator

Summation over context length allows our ICL regression score to be used for the comparison of tasks with significantly differing context lengths. An interpretation for each of different possible values of our ICL regression score is given in 2a. This approach builds off of Olsson et al.'s "ICL Score" [22] by generalizing their selection of 500 and 50 in-context examples and reducing along the context length, allowing for tasks with widely different context lengths to be directly compared. We list our baselines in Table 2b.

We replicate the baseline predictors for linear regression, sparse linear regression, and MLP regression from Garg et al. [6] due to the lack of a higher-performing baseline. However, we opted to use a pretrained GPT-2 model with identical structure to that used in Garg et al. to serve as a more calibrated baseline than Greedy Tree Learning or XGBoost. They showed superior decision tree ICL performance for a trained GPT-2 transformer compared to Greedy Tree Learning or XGBoost. For consistency with Park et al. [14] and due to the algorithmic hardness of `Sparse Parity`, we used our Mamba model trained on this task. Park et al. showed that Mamba can effectively learn this task, so we repeat our strategy as in `Decision Tree Regression` with our Mamba model (instead of GPT-2) as a baseline.

## 3.4 Reproducibility Statement

For ease of experimentation and reproducibility, we have built a typed, extensible, and modular Python codebase. We achieved this by identifying isolated processes in the training regime and

| Condition | Interpretation |
|---|---|
| $S_{\text{model}} > 1$ | model outperforms baseline |
| $S_{\text{model}} = 1$ | model matches baseline |
| $S_{\text{model}} < 1$ | model underperforms baseline |
| $S_{\text{model}} < 0$ | model underperforms zero estimator |

(a) Interpretation of possible $S_{\text{model}}$ values computed over context length.

| Task | Baseline Predictor |
|---|---|
| Linear | Least Squares |
| Sparse Linear | LASSO |
| MLP | 2-layer NN |
| Decision Tree | GPT-2 |
| Sparse Parity | Mamba |

(b) The baselines for each task. The 2-layer NN is trained for 1000 gradient steps, with a batch consisting of a randomly selected point in the context. GPT-2 and Mamba are trained for 500k steps on the specified task in the same format as all other models.

Figure 2: Predictors and conditions for computation and interpretation of ICL regression score.

structuring our code to reflect them. In particular, the specification of (1) a function class, (2) a model type, (3) an evaluation scheme, and (4) a stage of training under a curriculum are all inherent to the experiment archetype as proposed by Garg et al. [6] and repeated by others [8, 15, 14]. We integrate standard reporting software Weights and Biases [34] and leverage fast implementations of attention [35] and 1-D convolutions [36]. We also implement a configuration-based system for training, loading, and evaluating models to facilitate frictionless repeatability of all experiments.

# 4   Results

We confirm the results from Garg et al. [6] and Park et al. [14] that GPT-2 and Mamba can learn our first four regression tasks in context. Park et al. [14] that Mamba struggles to perform `Vector MQAR` while transformers and hybrid architectures excel. We note that Llama and GPT-2 have very comparable performance in `Sparse Parity` and `Vector MQAR`. We plot all qualitatively non-optimal squared error profiles in Figure 3 and all squared error profiles in Appendix B.

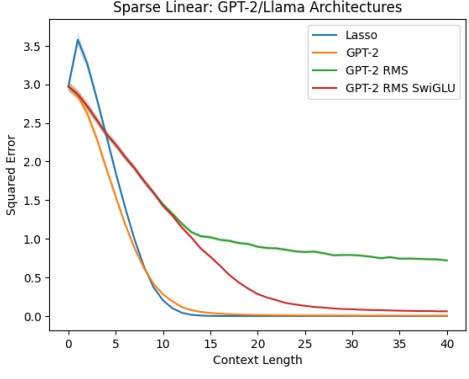
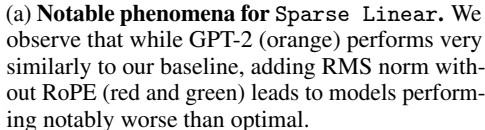

(a) **Notable phenomena for** `Sparse Linear`. We observe that while GPT-2 (orange) performs very similarly to our baseline, adding RMS norm without RoPE (red and green) leads to models performing notably worse than optimal.

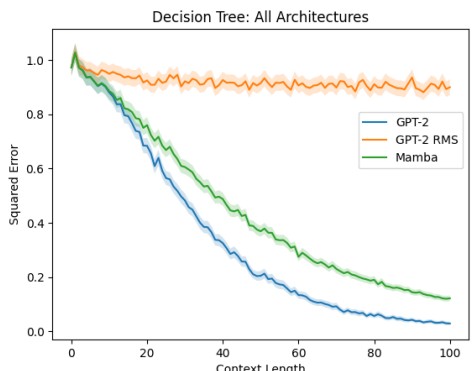

(b) **Notable phenomena for** `Decision Tree`. We note that Mamba (green) performs somewhat suboptimally while GPT-2 RMS (orange) fails to learn the task entirely.

Figure 3: Squared error profiles that do not exhibit near-optimal behavior. Shaded regions are 99% confidence intervals.

**Models can converge to suboptimal regression schemes.** We find that some model-task pairs produce suboptimal predictions, not as a result of insufficient training. A clear example is GPT-2 RMS SwiGLU (model 1.4) on `Sparse Linear`. This model appears to not achieve optimal error – achieving an ICL Regression Score of only $0.754$, opposed to $\sim 0.93$ by other models – and yet its performance does not significantly improve with more gradient steps. We plot the squared error achieved by various checkpoints for model 1.4 in Figure 4a. We observe that this error profile appears similar to that of models trained on the `Linear` task and so also examine the prediction quality of the

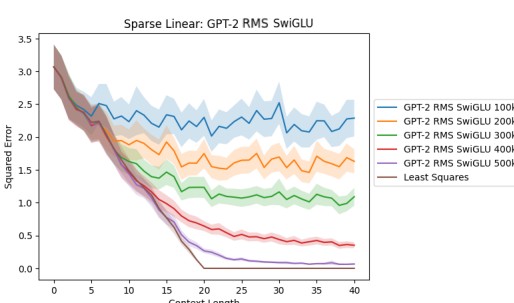

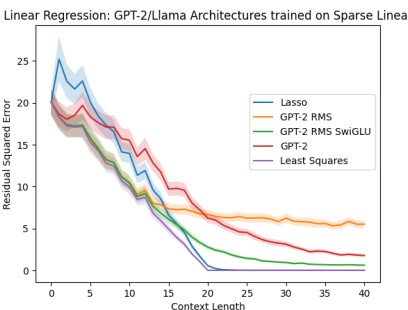

(a) **GPT-2 RMS SwiGLU Checkpoints on** `Sparse Linear.` We see that GPT-2 RMS SwiGLU converges to the least squares solution, despite Lasso being the optimal solution. This suggests that GPT-2 RMS SwiGLU fails to learn to utilize its context to its fullest extent.

(b) **GPT-2 RMS SwiGLU trained on** `Sparse Linear` **and evaluated on** `Linear.` When evaluated on a similar task to which it was trained on, GPT-2 RMS SwiGLU appears to perform *better* than its siblings, despite the fact that it performed *worse* than its siblings on its original task! This suggests that it learned a *different regression scheme* than GPT-2 on the same training data.

Figure 4: Detailing plots to showcase GPT-2 RMS SwiGLU (model 1.4) learning a more general but sub-optimal regression scheme when trained on `Sparse Linear`. Shaded regions are 99% confidence intervals.

same model (GPT-2 RMS SwiGLU trained on `Sparse Linear`) on `Linear` in Figure 4b. We find that it indeed mimics the error profile of least squares. This result builds on Akyürek et al.'s findings [19] in what functions transformer models develop representations of. Akyürek et al. analyzed algorithms representable by GPT-2 like architectures. We note that they did not examine other layer types such as Mamba Mixer or SwiGLU.

**Models can escape suboptimal regression schemes.** We see that GPT-2 SwiGLU (model 1.3) `Sparse Linear` on adopts a suboptimal regression scheme (least squares) partway in training, eventually unlearning its scheme in favor of the optimal regression scheme (lasso). We plot the squared error on Sparse Linear achieved by various checkpoints for Model 1.3 in Figure 5a, noting that the error of the checkpoint at 100k steps closely matches the error of least squares. Further, we examine the squared errors on Linear Regression for the various checkpoints for Model 1.3 in 5b and see that the checkpoint at 100k most closely matches least squares. This suggests that model 1.3 learned the linear regression scheme in the beginning of training, but was eventually able to learn to utilize the sparse nature of its training data.

**Models can fail to converge within our training horizon.** We find that a number of models performed strikingly poorly in their trained task. In particular, GPT-2 with Layer norm replaced by RMS norm (model 1.1) performed very poorly on `Sparse Linear Regression` and `Decision Tree`, as indicated by the lowest ICL Regression Score achieved in those tasks (0.535 and 0.114, respectively) and in Figures 3a and 3b. We also observe that GPT-2 with RMS and SwiGLU (model 1.4) also did not converge to a regression scheme, despite apparently modelling a different regression scheme entirely. Similarly, Mamba (model 3) did not converge to a training scheme on `Decision Tree` as illustrated in Figure 6a. We believe this suggests a lower training efficiency for certain architectures on these tasks.

**Models can fail to learn the task entirely.** In the case of `Decision Tree`, GPT-2 with RMS (model 1.1) failed to learn the task entirely as not only indicated by its final ICL Regression Score but also its consistency in achieving very high error throughout training. We plot squared error for various checkpoints in Figure 6b.

**ICL Regression Scores reflect qualitative information contained in squared-error plots.** Computed ICL Regression Scores are summarized in Table 3. Overall, most models are able to perform comparably to our baseline estimators, with nearly all examined models achieving a regression score of approximately 1 on all four function classes from Garg et al. (`Linear Regression`, `Sparse Linear Regression`, `2-Layer MLP`, `Decision Tree`). The ICL Regression Scores for `Linear`

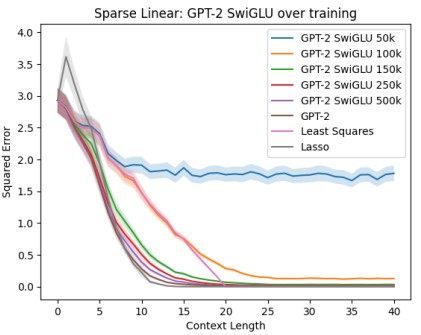
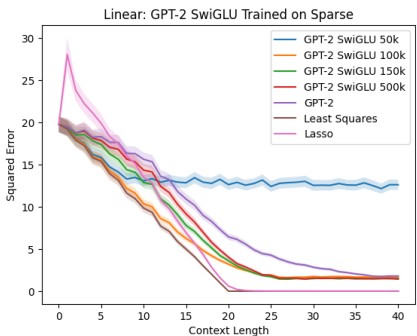

(a) **GPT-2 SwiGLU Checkpoints on** `Sparse Linear.` In the beginning of training, GPT-2 SwiGLU quickly converges to least squares, but it is able to escape this regression scheme and eventually has its error profile approach that of Lasso.

(b) **GPT-2 SwiGLU Checkpoints trained on** `Sparse Parity` **and evaluated on** `Linear Regression.` We see that an earlier checkpoint (100k) of GPT-2 SwiGLU outperforms later checkpoints on a similar task different from the task it was trained on.

Figure 5: Detailing plots to showcase GPT-2 SwiGLU (model 1.3) starting by learning a more general but sub-optimal regression scheme but eventually converging to the optimal regression scheme when trained on `Sparse Linear`. Shaded regions are 99% confidence intervals.

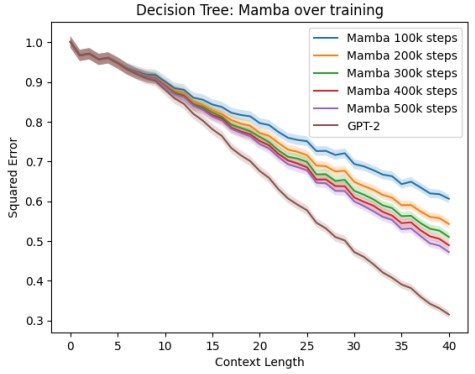
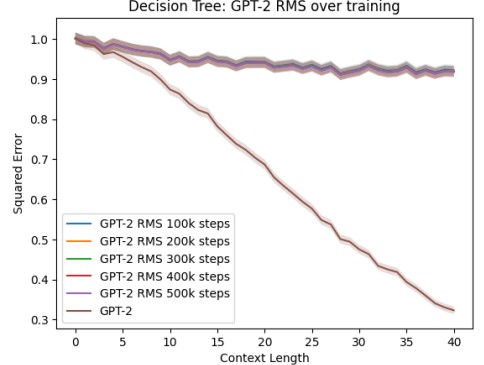

(a) **Mamba Checkpoints on** `Decision Tree.` We see that Mamba does keep improving its error profile throughout training. This suggests that Mamba did not reach convergence, and thus has lower training efficiency on this task.

(b) **GPT-2 RMS Checkpoints on** `Decision Tree.` We see that all checkpoints of GPT-2 perform very similarly, with little to no change in error profile throughout training.

Figure 6: Squared error as a function of context length computed for various checkpoints for both Mamba (model 3) and GPT-2 RMS (model 1.1) on `Decision Tree`. Shaded regions are 99% confidence intervals.

`Regression` and `2-Layer MLP`, along with their corresponding graphs of squared error as a function of context length, corroborate the claims from Garg et al. [6] that transformers can "learn" these tasks. Further, the ICL Regression Scores for `Sparse Parity` are consistent with Park et al. [14], with all hybrids between GPT-2, and Llama failing to "learn" the task and all hybrids between Llama and Mamba succeeding in "learning" the task. Indeed, the ICL Regression Score achieved by Mamba captures the qualitatively sub-optimal performance detailed above on `Decision Tree`.

## 5 Discussion

**Even simple function classes leave room for local minima.** We find that despite distilling down the phenomenon of In Context Learning to regression against simple function classes, there still exists room for models to adopt various regression schemes. This is supported by the apparent convergence

| | Model | Linear ±0.001 | Sparse Linear ±0.001 | 2-Layer MLP ±0.06 | Decision Tree ±0.001 | Sparse Parity ±0.001 |
|---|---|---|---|---|---|---|
| (1) | GPT-2 | 0.996 | 0.932 | 1.130 | 1.000* | 0.023 |
| (1.1) | GPT-2 RMS | 0.997 | 0.535 | 1.130 | 0.114 | – |
| (1.2) | GPT-2 RoPE | 0.995 | 0.927 | 1.130 | 1.004 | – |
| (1.3) | GPT-2 SwiGLU | 0.997 | 0.913 | 1.128 | 0.994 | – |
| (1.4) | GPT-2 RMS SwiGLU | 0.997 | 0.754 | 1.129 | 0.971 | – |
| (1.5) | GPT-2 RMS RoPE | 0.996 | 0.927 | 1.128 | 1.005 | – |
| (1.6) | GPT-2 RoPE SwiGLU | 0.996 | 0.929 | 1.129 | 1.011 | – |
| (2) | Llama | **0.997** | 0.933 | 1.129 | 1.007 | 0.023 |
| (2.1) | Llama RoPE-less | 0.996 | 0.928 | 1.130 | **1.018** | 1.000 |
| (2.2) | Llama SwiGLU-less | 0.996 | 0.927 | 1.129 | 0.980 | 1.000 |
| (2.3) | Llama RoPE,SwiGLU-less | 0.996 | **0.938** | 1.130 | 1.012 | 1.000 |
| (3) | Mamba | 0.995 | 0.925 | 1.123 | 0.832 | 1.000* |

Table 3: **ICL Regression Scores** for each architecture on each task, averaged over many sampled functions, with 95% confidence intervals in the headers for each row. Best-in-task values are in boldface except when not statistically significant from another architecture. GPT-2/Llama hybrids were not evaluated on Sparse Parity due to compute constraints and lack of supporting evidence that they should succeed. *These models were used as the baseline for this task.

of the error profiles of GPT-2 RMS (model 1.1) and GPT-2 RMS SwiGLU (model 1.4) to least squares regression for shorter context lengths.

**Hybrid architectures and function classes have varying levels of compatibility.** Specific hybrid architectures can hesitate to learn/converge for certain function classes. This behavior is especially apparent in GPT-2 RMS's (model 1.1) Decision Tree error graph and GPT-2 RMS SwiGLU's (model 1.4) Sparse Linear performance. It seems that GPT-2 RMS SwiGLU shows greater affinity towards learning least squares instead of LASSO. Certain hybrid architecture variations may place inductive biases on certain solution forms, resulting in extreme convergence times when these solution forms greatly vary from the optimal predictor's form.

**Extensible Research as Reproducible Research.** In the development of this work, continuously iterating to minimize the friction of reproduction has enabled rapid extension of our Python artifacts to support even abstractly defined *hybrid architectures*, which are often considered inextricable from highly bespoke code or dedicated packages such as xFormers [37]. We implore the reader to seriously consider the value of making their research extensible with a minimum of friction. We hope that our attempts to maximize extensibility and reproducibility contribute to the longevity of this work as a reliable, tested, and simple framework to use for studying simple function classes in context.

## 5.1 Limitations and Future Work

**We have only one training run performed on each model-task pair.** As a result, we have no estimation for how consistently observed phenomena appear with the given architectures. **We only train each model for a maximum of 500K steps.** Thus, when a model fails to converge within this window, we lose information on insightful trends that could possibly occur with further training.

**We do not empirically evaluate the effectiveness of ICL Regression Score or the usability of our provided code platform.** We compute no verifying metrics to establish how well ICL Regression Score generalizes or is robust to qualitatively distinct ICL regression tasks. Similarly, we perform no user study on the effectiveness of our code platform, presenting only our own experience.

**Future Work** In this paper we analyze ICL performance for GPT-2-Llama and Llama-Mamba hybrid architectures (9 total) on 6 tasks. Future relevant research could entail 1) expanding our architecture-space and streamlining our training-to-evaluation pipeline by creating an architecture search mechanism, 2) assessing our models on other sets of tasks, such as ones relating to language modeling or image classification, 3) verifying our results with additional training runs, 4) benchmarking model performance along hardware-related metrics.

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

 # A   Architectures

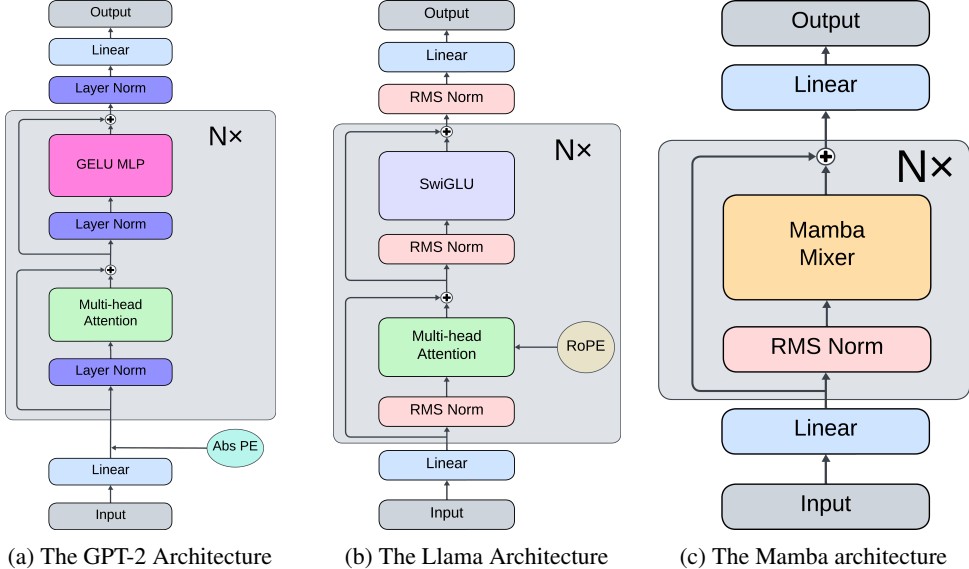

(a) The GPT-2 Architecture  (b) The Llama Architecture  (c) The Mamba architecture

Figure 7: The GPT-2, Llama, and Mamba architectures used in our regression tasks

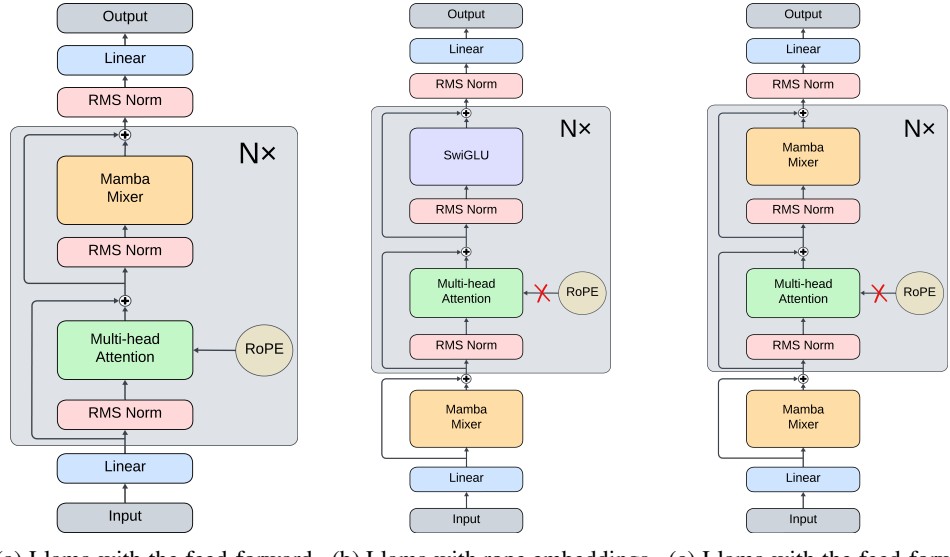

(a) Llama with the feed-forward block replaced by a Mamba Mixer block

(b) Llama with rope embeddings removed and a Mamba Mixer prepended to serve as a "positional embedder"

(c) Llama with the feed-forward block replaced by a Mamba Mixer block, rope embeddings removed, and a Mamba Mixer prepended to serve as a "positional embedder"

Figure 8: The hybrid architectures as modifications to Llama

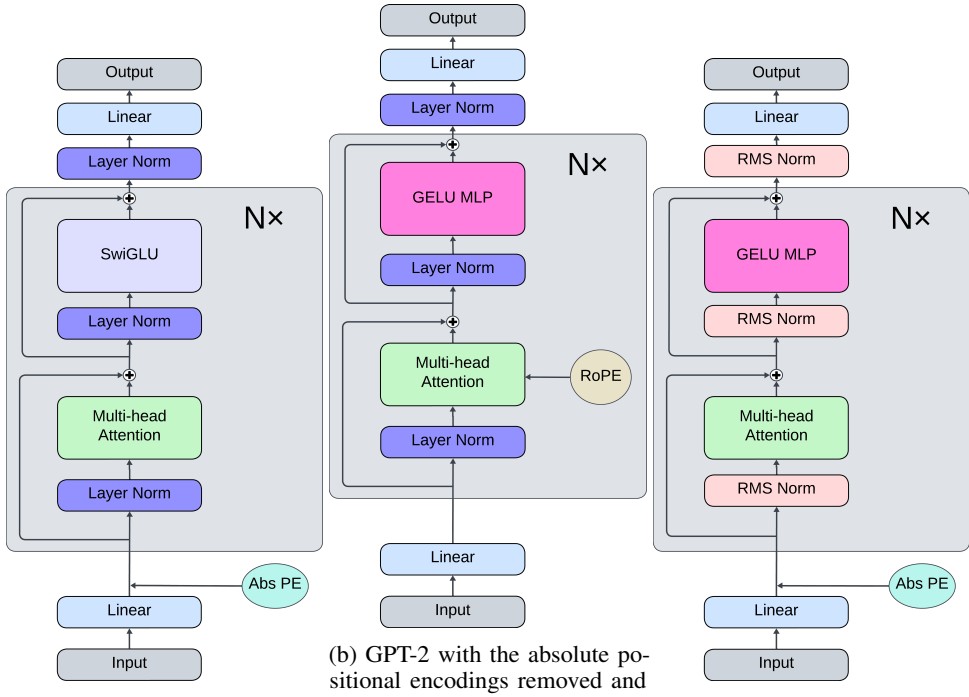

(a) GPT-2 with the GELU MLP replaced by a SwiGLU

(b) GPT-2 with the absolute positional encodings removed and rotary position embeddings included in attention

(c) GPT-2 with the Layer Norm replaced by an RMS Norm

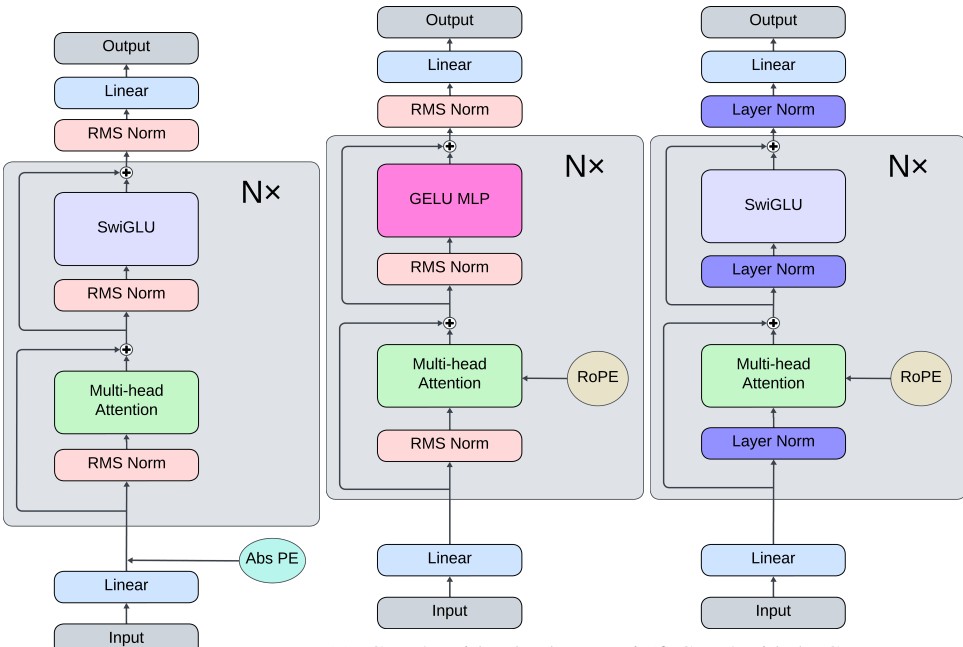

(d) GPT-2 with the GELU MLP replaced by a SwiGLU and the Layer Norm replaced by an RMS Norm

(e) GPT-2 with absolute positional encodings removed, rotary position embeddings included in attention, and the Layer Norm replaced by an RMS Norm

(f) GPT-2 with the GELU MLP replaced by a SwiGLU, absolute positional encodings removed, and rotary position embeddings included in attention

Figure 9: The hybrid architectures as modifications to GPT-2

# B Complete Experimental Results

## B.1 Linear Regression

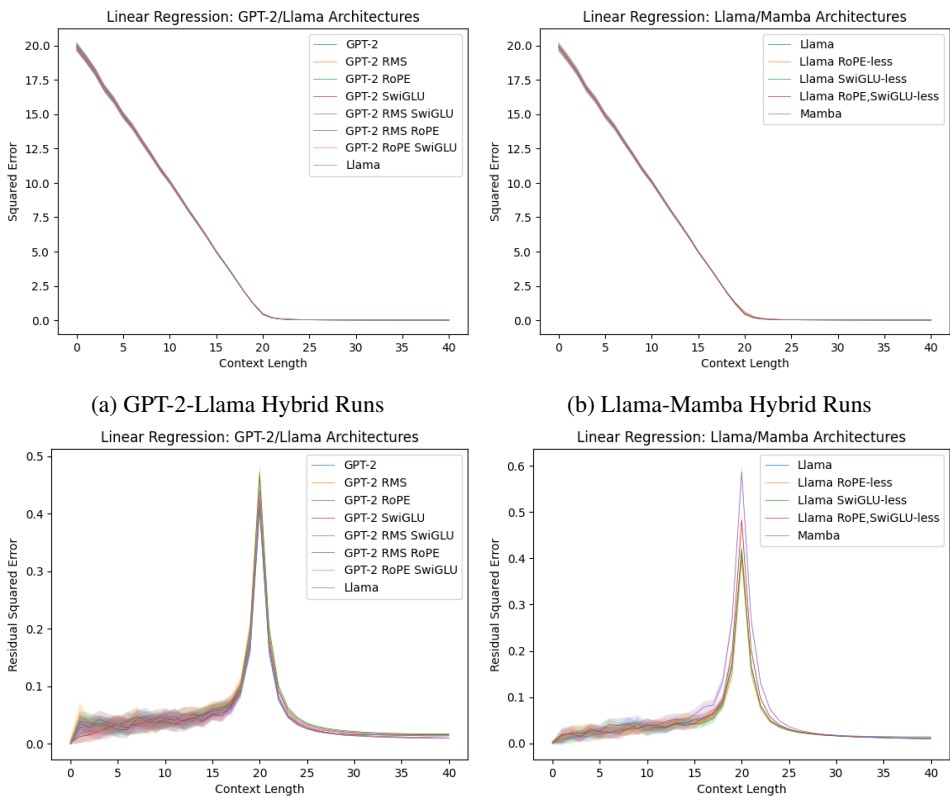

(a) GPT-2-Llama Hybrid Runs

(b) Llama-Mamba Hybrid Runs

(c) Residuals for GPT-2-Llama Hybrid Runs, taken against Least Squares

(d) Residuals for Llama-Mamba Hybrid Runs, taken against Least Squares

Figure 10: Linear Regression Runs with Residual Plots

## B.2 Sparse Linear Regression

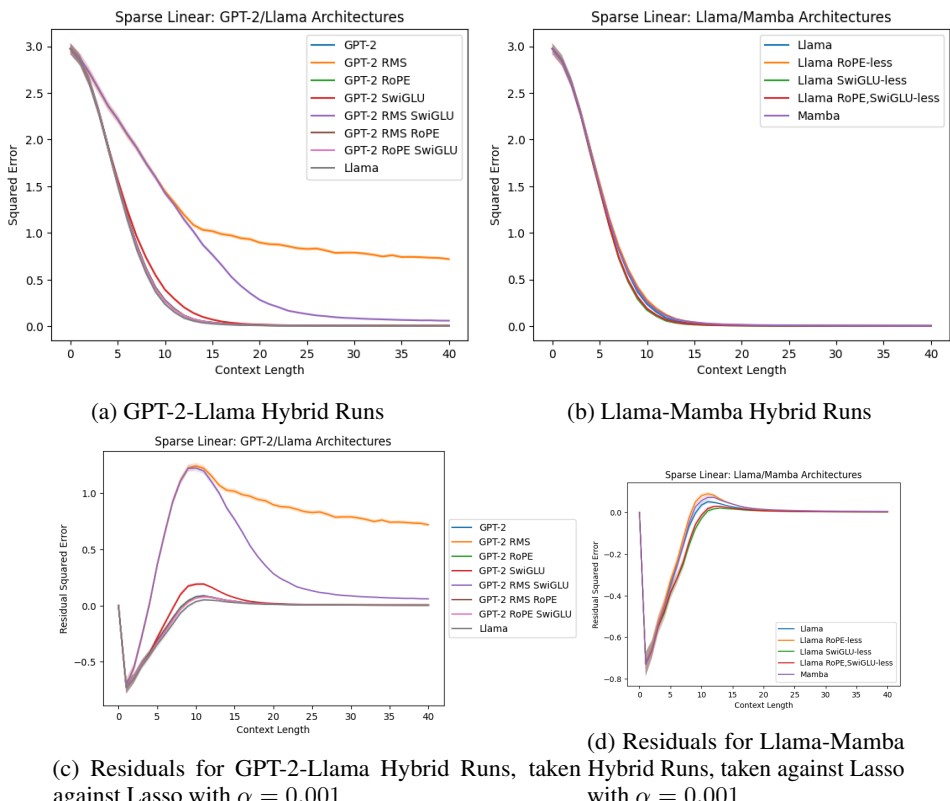

(a) GPT-2-Llama Hybrid Runs

(b) Llama-Mamba Hybrid Runs

(c) Residuals for GPT-2-Llama Hybrid Runs, taken against Lasso with $\alpha = 0.001$

(d) Residuals for Llama-Mamba Hybrid Runs, taken against Lasso with $\alpha = 0.001$

Figure 11: Sparse Linear Regression Runs

## B.3 Decision Trees

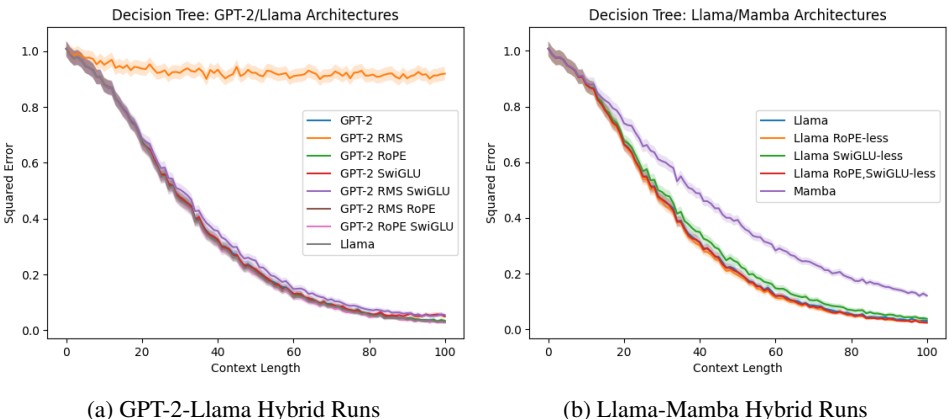

(a) GPT-2-Llama Hybrid Runs

(b) Llama-Mamba Hybrid Runs

Figure 12: Decision Tree Runs

## B.4 2-Layer NN Regression

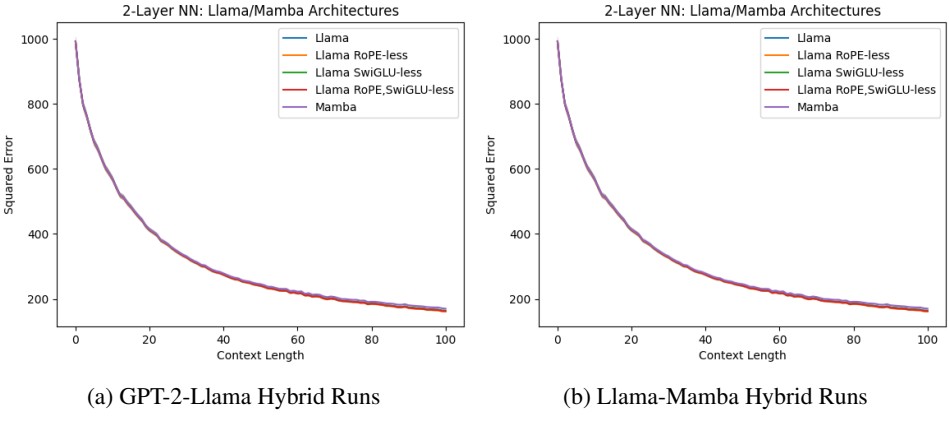

(a) GPT-2-Llama Hybrid Runs

(b) Llama-Mamba Hybrid Runs

Figure 13: 2-Layer NN Regression Runs

## B.5 Sparse Parity

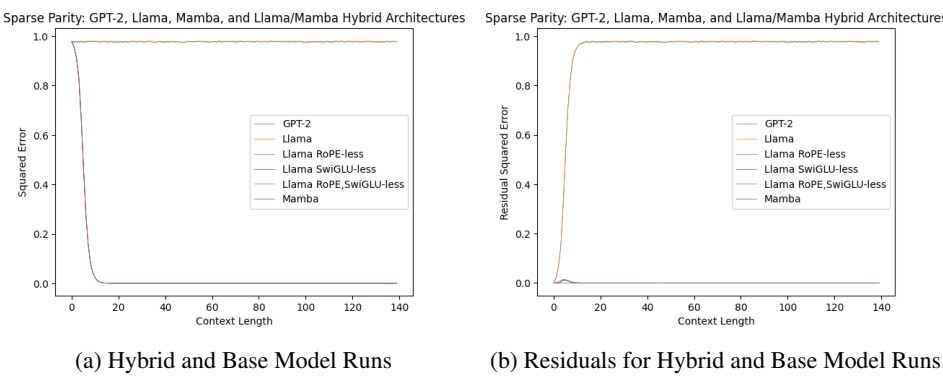

(a) Hybrid and Base Model Runs

(b) Residuals for Hybrid and Base Model Runs

Figure 14: Sparse Parity Runs with Residual Plots

## B.6 Vector MQAR

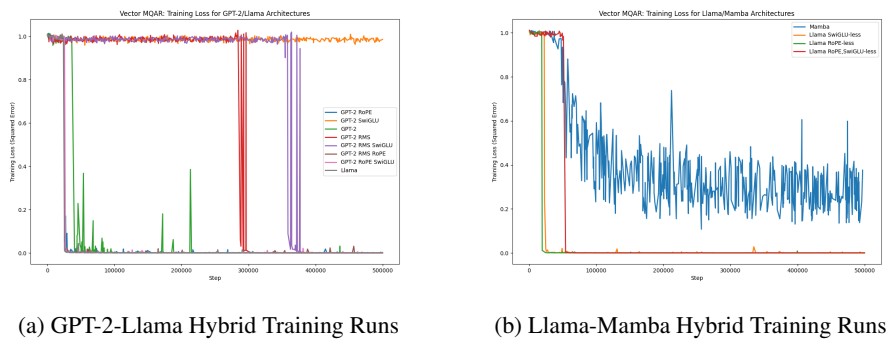

(a) GPT-2-Llama Hybrid Training Runs

(b) Llama-Mamba Hybrid Training Runs

Figure 15: Vector MQAR Training Runs

