# OpenReview forum: "Can Custom Models Learn In-Context? An Exploration of Hybrid Architecture Performance on In-Context Learning Tasks"
_NeurIPS.cc/2024/Conference — Submitted to NeurIPS 2024_

### Official Review · Reviewer_B8PQ · 2024-06-17

**Soundness:** 2
**Presentation:** 2
**Contribution:** 2
**Rating:** 3
**Confidence:** 3

**Summary:**

The authors explore a number of different attention-based architectures along with Mamba on a series of in-context regression tasks. The architectures vary in their choice of normalization, positional encodings, activations, and hybridization with Mamba. They discover some varying capacities for the different ICL tasks.

**Strengths:**

The paper appears to be closely related to [Park et al.](https://arxiv.org/abs/2402.04248), and I'm uncertain what additional scientific value it adds. As such, it's originality and significance appear to be limited.

**Weaknesses:**

As mentioned above, the present work seems to be closely related to [Park et al.](https://arxiv.org/abs/2402.04248), with little added insight. While an attempt has been made to explore minor architectural aspects like the choice of normalization, positional encodings, and activations, for the most part the changes seem to matter little (according to Table 3). I'm unsure what the key take-away is. Is there a particular architectural configuration that works best? What concrete practices can a user apply to improve their models' performance? The authors appear to have an ambition towards answering questions like these, but do not ultimately resolve them.

The results are sometimes difficult to interpret. At times, this is simply because the plots are unreadable, with text that is too small. I'm unsure how to interpret the in-context regression score. It looks like it's often <1 across models. Does this mean they all fail to outperform the baseline? Is this score comparable across different tasks?

A main objective of the paper is to compute numerics comparing models, but only one training run was executed for each experiment. If the compute budget is very limited, a more valuable approach may be to consider a smaller subset of models (e.g. keeping either GPT-2 or Llama, but not both) and simpler task parameterizations. Doing so will enable you to sweep across many more settings and increase your experiment replications, generating more convincing numerics.

Additional minor formatting comments:
- Line 25: consider compressing citations (e.g. with sort&compress)
- Table 1: third hline from the top intersects with text
- Consider plotting figures with point-markers for each data point, to clarify where exactly your data points fall
- For related models that vary by parameter (e.g. training iteraitons), consider using the same color but with different shadings / style
- Figure 15: text is unreadable
- Figure text overall is small and hard to read

**Questions:**

- Line 255 mentions that there is only a single run performed for each model-task pairs, yet many plots feature confidence intervals. Where do these confidence intervals come from?
- Why do you pick the model subsets that you do in constructing each plot? For instance, plot 3b has a title "Decision Tree: All Architectures" but only plots GPT-2 and Mamba (what about Llama, and the other variations?)
- Why apply Mamba only to Llama, and not to GPT-2? Since you mentioned compute was a bottleneck, why not keep either GPT-2 or Llama as a base model?
- How is the ICL regression score meant to be interpreted? It seems to depend strongly on the task, where some tasks will feature a score uniformly less than 1 across models, and greater than 1 for others.

**Limitations:**

The authors adequately state the limitations of their approach.

---

> ### Author Response · Authors · 2024-08-07
> **Author Response**
>
> We thank you for your thoughtful feedback and helpful comments. We briefly respond to some of the above questions and apologize for the otherwise low quality submission.
>
> The reviewer above highlights the limited scientific findings of the paper and proposes some avenues for uncovering some by building on the existing analysis.
>
> We indeed only perform a single training run for each model-task pair. We constructed confidence intervals from a distribution of errors achieved on 2048 batches of 64 functions and corresponding input points sampled from the task distributions (detailed in Table 1). This resulted in approximately 131k error values for each model at each context length, on which we assumed a normal distribution (due to similarity by visual inspection) and computed confidence intervals.
>
> In constructing each plot, we picked specific model subsets to increase readability. For instance, in plot 3b titled "Decision Tree: All Architectures," we only plot GPT-2, GPT-2 RMS, and Mamba. In all plotted tasks, we refrained from plotting more than one architecture that achieved near-optimal or otherwise identical error distributions.
>
> Mamba was applied only to Llama and not to GPT-2 to avoid repeating the contributions of Park et al. [1] addressing GPT-2/Mamba hybrid architectures. We analyzed hybrids between Llama and Mamba as representatives of attention-based and attention-free architectures because (to our knowledge) this was novel. We chose to also examine architectural “steps” between GPT-2 and Llama to verify that modern advancements in NLP are captured by our proposed regression score. We confirm this (admittedly loosely) by observing that native Llama obtains statistically similar or superior regression scores across all tasks.
>
> The proposed ICL regression score is intended to summarize the overall “performance” of a specific architecture on a certain task. It compares the predictive error of an architecture against that of a baseline. The ICL regression score depends strongly on the accuracy of our chosen baseline for each task and cannot be compared between tasks. For example, with a sub-optimal/poor baseline as used in the decision tree task, ICL regression scores will typically be higher. Conversely, for an information-optimal baseline such as in the linear regression or sparse linear regression tasks, regression scores will be strictly no greater than 1. Intuitively, the score should allow practitioners to quickly compare candidate architectures (such as GPT-2 RoPE SwiGLU vs Llama SwiGLU-less), but will hold little arithmetic value.
>
> We believe that the primary contributions of this paper are (1) the development of a codebase amenable to open-source contributions and (2) the creation of a score that can compare models within a synthetic regressive task. We thank the reviewer for clarifying that this is an obscured claim in the current draft.
>
> We once again thank you for the detailed review and informative suggestions.
>
> [1] Jongho Park, Jaeseung Park, Zheyang Xiong, Nayoung Lee, Jaewoong Cho, Samet Oymak, Kangwook Lee, and Dimitris Papailiopoulos. Can mamba learn how to learn? a comparative study on in-context learning tasks. arXiv preprint arXiv:2402.04248, 2024.

---

> > ### Comment · Reviewer_B8PQ · 2024-08-10
> >
> > Thank you for your detailed rebuttals and clarifying comments. After reading the other reviews and rebuttals, I concur with the other reviewers in that 1) the impact of these experiments remains unclear and 2) the present manuscript seems to be in a rough, unfinished state. For these reasons, I retain my current score.

---

### Official Review · Reviewer_y95k · 2024-07-13

**Soundness:** 3
**Presentation:** 1
**Contribution:** 2
**Rating:** 4
**Confidence:** 4

**Summary:**

The authors build on the ICL work of [1] and [2], wherein networks are "trained to in-context learn" several tasks of varying complexity (e.g., Linear Regression, Sparse Linear Regression, Vector MQAR, etc.).  Carrying on from [2], the main contribution of the presented work is the combination of various permutations of the attention/SSM blocks from GPT-2, LLama-2, and Mamba models.  Such permutations include swapping the Layer Norm in GPT-2 attention blocks with RMSNorm, the LLama-2 SwigGLU with a Mamba Mixer, and so on; in total, 9 different hybrid combinations are considered.  For each task and specific hybrid model, the model is trained over task samples for 500k steps, then evaluated on ICL performance on that task.  In addition to reporting squared error per task, the authors also propose a new metric, called the "ICL regression score."  The results of specific model-task pairs are plotted and several trends are discussed.

[1] Shivam Garg, Dimitris Tsipras, Percy S Liang, and Gregory Valiant. What can transformers learn in-context? a case study of simple function classes. Advances in Neural Information Processing Systems, 35:30583–30598, 2022.
[2] Jongho Park, Jaeseung Park, Zheyang Xiong, Nayoung Lee, Jaewoong Cho, Samet Oymak, Kangwook Lee, and Dimitris Papailiopoulos. Can mamba learn how to learn? a comparative study on in-context learning tasks. arXiv preprint arXiv:2402.04248, 2024.

**Strengths:**

**Originality** - while the authors largely adapt the "trained to in-context learn" framework from other works, they consider new model configurations.  This can be useful to determine what permutations of model components leads to failure modes in ICL.

**Quality** - In considering 9 different model configurations for 8 different tasks, the author thoroughly consider a large number (72) of different LLM building blocks and their ICL capabilities across tasks of varying complexities.

**Significance** - The framework of [1] has grown as an interesting alternative to standard ICL, as a quick means to assess the ICL success and failure modes of different models.  Thus, the presented work can aid in flushing out this information over new LLMs and previously proposed tasks.

[1] Shivam Garg, Dimitris Tsipras, Percy S Liang, and Gregory Valiant. What can transformers learn in-context? a case study of simple function classes. Advances in Neural Information Processing Systems, 35:30583–30598, 2022.

**Weaknesses:**

### Clarity
There is significant room to improve the clarity of the presented work.  Currently, several important details are missing (discussed below), key concepts are not fully explained, and the current paper could benefit from an editorial pass (it is currently difficult to read).  It is also important to note that, while the authors presented the results of model + task pairs and detailed where models failed, no possible explanations and follow up ablation studies were conducted; the question of "why" remains for all the presented results.  E.g.,
> Specific hybrid architectures can hesitate to learn/converge for certain function classes.

Do the authors have intuition or an explanation which can be explored to explain this?  Arguably, this is one of the most important contributions to be made in a large empirical review like the presented paper, i.e., to make sense of the experiments to gain a greater intuition for why an LLM behaves in a certain way.

Wrt important missing information:
- "We replicate the function classes Linear Regression, Sparse Linear Regression, 2-Layer MLP Regression, and Decision Tree Regression from Garg et al. [6] as they present a wide range of "difficulty" for sequence models. In addition, to capture the existence of some ICL ability, we also regress onto the two function classes examined in Park et al. [14]: parity function with induced sparsity (Sparse Parity) and parallel associative recall (Vector MQAR)." <- - How training instances are produced per task? How many test samples are produced per task?  If this follows Garg et al., then each model is trained *from scratch* on 40 samples per task. Can you please clarify and state these in the main text?
- It is not clear	what the author's mean by "zero estimator."  Is this the zero shot prediction? Correspondingly, it is not clear exactly what the presented ICL regression score represents.
- "To determine task-specific ICL ability, our sequence models regress onto the functions shown above [14]."  <- It would help to clearly state the paper trains the models "from scratch" to in-context learn, as in previous works.

**Questions:**

See weaknesses.

**Limitations:**

The did well to state limitations of the presented evaluation.

---

> ### Author Response · Authors · 2024-08-07
> **Author Response**
>
> We appreciate your diligent feedback and practical critique. We attempt to clarify some details missed by the submission below.
>
> We trained all checkpoints used in our analysis from scratch on synthetic data generated by sampling parametrized functions and x values from the distributions specified in Table 1. Those sampled x values and functions are then collected into prompts that follow the structure described in Section 3. We will add this with more clarity to the main text.
>
> The “zero estimator” predicts 0 for all regression targets, which in all tasks is the estimate with minimal expected squared error conditioned on *no* context.
>
> The ICL regression score aims to summarize the overall “performance” of a specific architecture on a certain task. It compares the predictive error of an architecture against that of a baseline. ICL regression score strongly depends on the accuracy of our chosen baseline for each task and consequently cannot be compared between tasks. For example, with a poor “baseline” as we use in the decision tree task, ICL regression scores will typically be higher. Conversely, for an information-optimal baseline such as in the linear regression task, regression scores will be strictly no greater than 1. Intuitively, we suggest treating it as a “score” achieved by an architecture on a task – one that holds little arithmetic value, but can allow practitioners to quickly compare candidate architectures.
>
> We would like to reiterate that we are grateful for the reviewer’s genuine and constructive feedback.

---

> > ### Comment · Reviewer_y95k · 2024-08-11
> > **Acknowledgement of rebuttal**
> >
> > I thank the authors for their response.  I've read the other reviews and agree the submission looks to be in an unfinished state and the presented experiments need further exploration to better distinguish from previous work (as well as articulate what any major takeaways are).  I am retaining my score and wish the authors the best in future revisions.

---

### Official Review · Reviewer_q9Lv · 2024-07-14

**Soundness:** 2
**Presentation:** 3
**Contribution:** 2
**Rating:** 4
**Confidence:** 5

**Summary:**

This work presents a codebase for benchmarking the in-context learning ability of language models, especially for hybrid models. In addition, several empirical results are presented to show that some model architectures fail entirely or have suboptimal performance on specific in-context learning tasks.

**Strengths:**

* The codebase for studying in-context learning ability could be useful to understand capabilities and limitations of hybrid models, which will accelerate research in this area.
* It is interesting to find that even a small change in architecture (e.g., adding RMS to GPT-2) will lead to noticeable differences on some tasks (e.g., sparse linear). It would be interesting to investigate the root reason behind that.

**Weaknesses:**

* I feel that it is hard to assess the contribution of this work. It seems that this work's main contribution is the implementation of the in-context learning ability benchmark codebase. While such a codebase is important and useful, I did not find what technical challenges the codebase is trying to address and the effectiveness of the codebase.
* Another contribution is the empirical findings of the relationship between architectures and per-task performance on in-context learning. However, I found the empirical results are not systematic and hard to interpret. I am unsure how these findings motivate future architecture design.

**Questions:**

As stated above, what's the main contribution/innovation of the codebase? What conclusion/takeaway/intuition can we learn from the empirical results?

**Limitations:**

see above

---

> ### Author Response · Authors · 2024-08-07
> **Author Response**
>
> We thank you for the earnest review and encouraging feedback. We otherwise apologize for the unpolished writing.
>
> The reviewer above identifies the primary contribution of this paper to be the associated code but finds that the empirical analysis is lacking systematism, motivation, and a digestible upshot.
>
> We cede that the paper does not explicitly present a digestible result for practitioners, but we claim that the ICL regression score we detailed can serve as a quick-and-dirty tool for comparing the ICL ability of architectural variations.
>
> As for our code, we improved the modularity and implementation of abstractions over existing codebases such as in Garg et al. [1], Park et al. [2], and Lee et al. [3]. We allowed for extensions to architecture, benchmarking, and even training schemes to occur often independently and simultaneously without causing conflicts in standard version control systems like Git. A particularly illustrative example of this is our yaml-only hybrid architecture specification, which was developed concurrently with our evaluation tooling. We also used best practices in open-source repository management such as continuous integration, behavioral testing, and reproducible development and deployment environments.
>
> Again, we thank you for the sincere review.
>
> [1] Garg S, Tsipras D, Liang PS, Valiant G. What can transformers learn in-context? a case study of simple function classes. Advances in Neural Information Processing Systems. 2022.
>
> [2] Jongho Park, Jaeseung Park, Zheyang Xiong, Nayoung Lee, Jaewoong Cho, Samet Oymak, Kangwook Lee, and Dimitris Papailiopoulos. Can mamba learn how to learn? a comparative study on in-context learning tasks. arXiv preprint arXiv:2402.04248, 2024.
>
> [3] Ivan Lee, Nan Jiang, and Taylor Berg-Kirkpatrick. Is attention required for ICL? Exploring the Relationship Between Model Architecture and In-Context Learning Ability. arXiv preprint arXiv:2310.08049, 2023.

---

### Official Review · Reviewer_xju7 · 2024-07-29

**Soundness:** 3
**Presentation:** 2
**Contribution:** 1
**Rating:** 4
**Confidence:** 3

**Summary:**

This work presents an analysis of in-context learning (ICL) for a variety of hybrid architectures (composed of different blocks from preexisting large language model architectures) on different regression tasks. The experiments are built on top of a couple of prior works [1, 2] that also explored ICL in similar contexts. This paper highlights that several prior results can be reproduced, and for novel hybrid architectures which are the main focus of this work – most of them converge to optimal solutions while some others can escape suboptimal solutions or even fail to converge in the first place. The authors also propose a new metric “ICL regression score” to evaluate ICL performance in comparison to a known baseline. The modularized code for this work is publicly available for the broader scientific community.


[1] Garg S, Tsipras D, Liang PS, Valiant G. What can transformers learn in-context? a case study of simple function classes. Advances in Neural Information Processing Systems. 2022.
[2] Park J, Park J, Xiong Z, Lee N, Cho J, Oymak S, Lee K, Papailiopoulos D. Can mamba learn how to learn? a comparative study on in-context learning tasks. arXiv preprint arXiv:2402.04248. 2024.

**Strengths:**

- The open source codebase with simple abstractions and interfaces to facilitate reproducibility, extensions, and modifications are a welcome contribution.
- The intuitive explanation behind the ICL regression score values in Figure 2(a) are well-appreciated and helpful to follow along the results.
- The authors evaluate multiple architectures and tasks and clearly outline what components they are using from prior works to build on top of.

**Weaknesses:**

- The paper is not very well-motivated. Why are hybrid architectures (especially the two that are focused on) important to study? What intuitions or profound reasons drive the authors to make the experimental design choices that they did?
- Additionally, use cases for ICL itself are not well-motivated. Are there any practical use cases that warrant such extensive evaluation? The writing is not easy to understand for a reader not very up-to-date with the ICL literature.
- The technical novelty of the work is limited.
- The results are presented in a manner where the performance metrics are reported for the 12 architectures and 5 tasks but it is not very clear what the reader or the scientific community working on ICL should take away from the results. Are there patterns regarding why certain hybrid architecture + task combinations make ICL shine compared to the baselines and why some others do not? A lot of the interpretation of such results is left to the reader to figure out. A lack of a deeper understanding and intuition about the reasons behind the results makes it hard to see solid/impactful takeaways that others could build on top of.
- The authors mention and describe a 6th task Vector MQAR but do not report or discuss any of its results in detail in the main text of the paper. One figure is present in the Appendix but it is not explained and it is too hard to read the text in the figure.
- Some typos:
     - Line 161: Mention the word “**Figure**” before 2a.
     - Line 164: “**Figure 2b**” instead of “Table 2b”
     - Line 185: Park et al. [14] **show** that ..
     - Line 202: “Sparse Linear ~~on~~ adopts a suboptimal..”
- References to result tables (Table 3 for lines 213, 230)  and model descriptions (Figure 1a for lines 194, 204, 201, 16 etc.) could further enhance readability and the user’s understanding.

**Questions:**

- What is the main motivation behind studying hybrid architectures for ICL? For a reader not well-versed with the ICL literature, it is not clear from the paper.
- Why are the specific hybrid architectures chosen (GPT2-Llama and Llama-Mamba)? Is it only because no one has studied them before or are there other profound reasons why these could be of interest to study?
- On line 35, what is meant by “richly benchmarking”?
- Could you briefly describe what kind of structural changes were necessary to the codebase of Garg et al.? A general understanding will make it more clear why the codebase itself is a major contribution as listed currently in the draft.
- Lines 60-61: What optimal train loss is being referred to here? There is no loss associated with ICL itself, right (as that only consists of in-context examples/demonstrations)? Does this loss refer to the ground truth functions being learned or the baselines?
- Line 80: which models are being referred to here? There is no “real training” during ICL itself. Are the functions/regression targets being referred to here or the baselines? The terminology used in the paper should be clarified.
- It is not clear what data the 12 hybrid architectures are trained on. Do the authors train them from scratch? If yes, what datasets are used? Else are pretrained weights used for the different blocks? Are the models finetuned? Something else?
- Why aren’t the baselines and detailed results for Vector MQAR discussed in the paper draft? Why does Mamba struggle with MQAR but does pretty well on all other 5 tasks?
- Why are some hybrid architectures converging to suboptimal regression schemes versus others are not? Do you have some intuition behind this finding?
- Similarly, why do some models escape suboptimal regression schemes while others fail to converge entirely?
- For Fig 6(a), could you try training Mamba checkpoints longer than 500k checkpoints to verify if it actually converges to the baseline (GPT-2’s performance)? Were these checkpoints trained by the authors themselves or used from a pretrained model? What data was used for training?
- What do the authors mean by “context length” in most of their figures? Does it refer to the number of ICL examples or the number of prompt tokens? Can the authors share a few examples of sample prompts that were used for ICL with varying context lengths?
- In Figure 6(b), what about GPT2-RMS might explain its inability to learn the decision tree function while all other hybrid architectures used do pretty well? Why is the RMS norm as the distinguishing factor for this hybrid architecture detrimental to the decision tree and sparse linear tasks but not others?
- In lines 244-246, the authors mention that “certain hybrid architecture variations may place inductive biases on certain solution forms, resulting in convergence times when these solution forms greatly vary from the optimal predictor’s form”. Can the authors give some examples of such inductive biases for some architectures studied in this work?
- Why are some values in the last column of Table 3 not filled in?

**Limitations:**

In terms of negative societal impacts, could the potential misuse of natural language tasks via the hybrid architectures presented be a legitimate concern? It is certainly out of scope of this work for evaluation but could be listed as part of the broader impacts section of the checklist.

---

> ### Author Response · Authors · 2024-08-07
> **Author Response (part 1/2)**
>
> We are grateful for your detailed feedback and thorough critique of the submission. We provide a clarification to questions about motivating the submission in this response and reserve responses to specific questions for another response. We apologize for the poor submission.
>
> The reviewer above details limited motivations, novelty, and clarity of the paper, while mentioning the potential usefulness of an improved codebase.
>
> Leading language models have ballooning inference costs, and finding a deeper correlation between architectural sub-blocks and ICL capacities can help inform designing for the cost/performance trade-off at inference time. Since in-context learning is the primary avenue by which LLMs are useful to a wide range of tasks, we propose that it is then useful to gauge the ICL ability conferred by small architectural choices. To this end, we contend that since there exists some synthetic ICL regression/classification tasks with varying performance across architectures, it is reasonable to assume that there is a set of benchmarks that can capture different qualitative capacities that we as humans intuitively aggregate into “ICL ability”. As a result, performance on such a set of benchmarks should be indicative of ICL capacities in language models. We referred to a suite of benchmarks that approach this comprehensiveness as “richly benchmarking” on line 35.
>
> Existing work on synthetic ICL tasks such as those in the paper examine either GPT-2 or attention-free architectures (such as Mamba in Park et al. [1], a wide array of SSMs and recurrent models in Lee et al. [2]). The only exception we are aware of to this is the inclusion of Llama in the survey performed by Lee et al. [2]. As a result, we selected Llama and Mamba as architectures representative of current work in NLP for both attention-based and attention-free architectures and chose mixes between them to examine the change (if any) in the ICL capacities of both attention-based architectures produced by recent advances and hybrid attention architectures.
>
> Our changes to the codebase were primarily logistical, but constitute an overhaul from repositories that build directly from Garg et al. [3]. We improved distinctions between abstractions and their implementations, enabling (1) extensions to model architecture, evaluation techniques, and even training schemes to occur often independently and simultaneously without causing version control conflicts and (2) best practices in open-source repository management such as continuous integration, behavioral testing, and reproducible development/deployment environments. We contend that this is a necessary contribution to facilitate examination of such small architectural variations in this paper and future work.
>
> [1] Jongho Park, Jaeseung Park, Zheyang Xiong, Nayoung Lee, Jaewoong Cho, Samet Oymak, Kangwook Lee, and Dimitris Papailiopoulos. Can mamba learn how to learn? a comparative study on in-context learning tasks. arXiv preprint arXiv:2402.04248, 2024.
>
> [2] Ivan Lee, Nan Jiang, and Taylor Berg-Kirkpatrick. Is attention required for ICL? Exploring the Relationship Between Model Architecture and In-Context Learning Ability. arXiv preprint arXiv:2310.08049, 2023.
>
> [3] Garg S, Tsipras D, Liang PS, Valiant G. What can transformers learn in-context? a case study of simple function classes. Advances in Neural Information Processing Systems. 2022.

---

> ### Author Response · Authors · 2024-08-07
> **Author Response (part 2/2)**
>
> We respond to clarifying questions in this response.
>
> We trained all models from scratch, using standard initializations. The training data is synthetic: randomly sampled vectors $x_i$ for each in-context example and only one set of parameters $ \theta $ for each sequence arranged as illustrated in Section 3. The distributions are summarized in Table 1. For example, a batch of $8$ sequences with 40 in-context examples each results in $8 \times 40 =  320$ sampled $x$ values and $8$ sampled sets of parameters ($ \theta $).
>
> We conflate “context length” with the number of $ \( x, f(x) \) $ pairs (i.e. in-context examples). This makes the number of prompt tokens simply one more than double the “context length”. This is because in-context examples must have both an input value $x$ and label value $ f(x) $ *and* the prompt must have a query token of an unlabeled input value $ x_{query} $. See the beginning of Section 3 for a (hopefully helpful) visual.
>
> For example, consider a linear function of input dimension 3 (we use 20 in the paper) and weights [1, 1, 1], producing a prompt of “context length” 2:
>
> $$ \\begin{bmatrix} 1 \\\\ 2 \\\\ 1 \\end{bmatrix}, \\begin{bmatrix} 4 \\\\ 0 \\\\ 0 \\end{bmatrix}, \\begin{bmatrix} -1 \\\\ 1 \\\\ 0.5 \\end{bmatrix}, \\begin{bmatrix} 0.5 \\\\ 0 \\\\ 0 \\end{bmatrix}, \\begin{bmatrix} 0 \\\\ 0.2 \\\\ 1 \\end{bmatrix} $$
>
> In this case, the regression target is 1.2.
>
> Another example: consider a sparse linear function of input dimension 3 and weights [0, 1, 0], producing a prompt of “context length” 3:
>
> $$ \\begin{bmatrix} 0.6 \\\\ 0.4 \\\\ -0.2 \\end{bmatrix}, \\begin{bmatrix} 0.4 \\\\ 0 \\\\ 0 \\end{bmatrix}, \\begin{bmatrix} -0.3 \\\\ 0.8 \\\\ -0.1 \\end{bmatrix}, \\begin{bmatrix} 0.8 \\\\ 0 \\\\ 0 \\end{bmatrix}, \\begin{bmatrix} 1 \\\\ -1.4 \\\\ 0.4 \\end{bmatrix}, \\begin{bmatrix} -1.4 \\\\ 0 \\\\ 0 \\end{bmatrix}, \\begin{bmatrix} 0.1 \\\\ -0.8 \\\\ 1.1 \\end{bmatrix} $$
>
> In this case, the regression target is -0.8.
>
> ---
>
> Where we say “optimal loss,” there are closed-form predictors that achieve theoretically minimal (average) loss at each context length. For example, optimal loss at linear regression with squared-error loss is that of the least squares regression line (as opposed to 0 error at all context lengths, which is the ground truth). As far as the authors are aware, regressing to a random 2-layer MLP or depth 4 decision tree does not have an optimal scheme, so we use the more general term “baselines” to refer to regression schemes that serve as points of comparison.
>
> Line 80 refers to transformer-/mamba-like models. We defined our training loss to be the average squared error between the model’s prediction $ \hat f_{\theta}(x_{query}) $ and the ground truth $ f_{\theta}(x_{query}) $. Thus our training scheme was to minimize expected loss over many sampled functions.
>
> We continued training Mamba up to 1 million gradient steps and found that it continued to (very slowly) approach the regression scheme achieved by GPT-2. As of this posting, our 1.1 million step checkpoint only continues this trend, but does not yet reach the same error profile.
>
> The last column of Table 3 contains unfilled entries because we inferred that since both GPT-2 and Llama failed entirely at sparse parity, that hybrid architectures would perform similarly. As a result, we omit those model-task pairs.
>
> The form of the Vector MQAR we replicated did not consider querying models at multiple context lengths, so using any baseline to compute a regression score would reduce to multiplication by a scalar value. This makes formulating a baseline for Vector MQAR not meaningful in our context. We did not discuss detailed results for Vector MQAR as our results almost exactly matched direct extrapolations from the results of Park et al. We will expand discussion of these results in the paper.
>
> We do not have a definitive answer for why Mamba only struggles with MQAR. Park et al. [1] propose that the latent representation space expressible by Mamba for this task (a vector of dimension 512 per layer) is not expressive enough for this task (which calls for storing 128 pairs of 20-dimensional vectors, or 5120 real values) and becomes “overwhelmed” with long contexts. We generally agree with this, and also affirm that any recurrent model’s latent space along the sequence length can be saturated this way. To clarify, the models trained on this task are smaller than those from other tasks to ensure that the attention-based models converged, suggesting that computing a gradient from only a single regression target per sequence limits the complexity of models that are trainable on this task.
>
> Again, we thank you for the complete review and helpful guidance.
>
> [1] Jongho Park, Jaeseung Park, Zheyang Xiong, Nayoung Lee, Jaewoong Cho, Samet Oymak, Kangwook Lee, and Dimitris Papailiopoulos. Can mamba learn how to learn? a comparative study on in-context learning tasks. arXiv preprint arXiv:2402.04248, 2024.

---

> > ### Comment · Reviewer_xju7 · 2024-08-14
> > **Acknowledgement of rebuttal**
> >
> > I thank the authors for the detailed answers to all my questions. The responses presented in the rebuttal can greatly improve the readability of the draft. I agree with other reviewers that the paper has further scope for improvement. I will thus retain my score and wish the authors the best in preparing a revised version of this work.

---

### Author Response · Authors · 2024-08-07
**Clarification to motivations and contributions**

We deeply thank the reviewers for their thorough and thoughtful responses. We apologize for the rough writing and late rebuttal. We understand that this may not be read, but we would like to have this collected with this submission for future reference.

Leading language models have ballooning inference costs, and finding a deeper correlation between architectural sub-blocks and ICL capacities can help inform designing for the cost/performance trade-off at inference time. Since in-context learning is the primary avenue by which LLMs are useful to a wide range of tasks, we propose that it is then useful to gauge the ICL ability conferred by small architectural choices. To this end, we contend that since there exists some synthetic ICL regression/classification tasks with varying performance across architectures, it is reasonable to assume that there is a set of benchmarks that can capture different qualitative capacities that we as humans intuitively aggregate into “ICL ability”. As a result, performance on such a set of benchmarks should be indicative of ICL capacities in language models. We referred to a suite of benchmarks that approach this comprehensiveness as “richly benchmarking” on line 35.

Existing work on synthetic ICL tasks such as those in the paper examine either GPT-2 or attention-free architectures (such as Mamba in Park et al. [1], a wide array of SSMs and recurrent models in Lee et al. [2]). The only exception we are aware of to this is the inclusion of Llama in the survey performed by Lee et al. [2]. As a result, we selected Llama and Mamba as architectures representative of current work in NLP for both attention-based and attention-free architectures and chose mixes between them to examine the change (if any) in the ICL capacities of both attention-based architectures produced by recent advances and hybrid attention architectures.

We propose an ICL regression score to summarize the overall “performance” of a specific architecture on a certain task. It compares the predictive error of an architecture against that of a baseline. The ICL regression score depends strongly on the accuracy of our chosen baseline for each task and cannot be compared between tasks. For example, with a sub-optimal/poor baseline as used in the decision tree task, ICL regression scores will typically be higher. Conversely, for an information-optimal baseline such as in the linear regression or sparse linear regression tasks, regression scores will be strictly no greater than 1. This score can serve as a quick-and-dirty tool for comparing the ICL capacities of candidate architectures.

Our changes to the codebase were primarily logistical, but constitute an overhaul from repositories that build directly from Garg et al. [3]. We improved distinctions between abstractions and their implementations, enabling (1) extensions to model architecture, evaluation techniques, and even training schemes to occur often independently and simultaneously without causing version control conflicts and (2) best practices in open-source repository management such as continuous integration, behavioral testing, and reproducible development/deployment environments. We contend that this is a necessary contribution to facilitate examination of such small architectural variations in this paper and future work.

We would like to again thank the reviewers for their time and consideration.

[1] Jongho Park, Jaeseung Park, Zheyang Xiong, Nayoung Lee, Jaewoong Cho, Samet Oymak, Kangwook Lee, and Dimitris Papailiopoulos. Can mamba learn how to learn? a comparative study on in-context learning tasks. arXiv preprint arXiv:2402.04248, 2024.

[2] Ivan Lee, Nan Jiang, and Taylor Berg-Kirkpatrick. Is attention required for ICL? Exploring the Relationship Between Model Architecture and In-Context Learning Ability. arXiv preprint arXiv:2310.08049, 2023.

[3] Garg S, Tsipras D, Liang PS, Valiant G. What can transformers learn in-context? a case study of simple function classes. Advances in Neural Information Processing Systems. 2022.

---

### Decision · Program_Chairs · 2024-09-25

**Decision:**

Reject

**Comment:**

The paper presents an empirical study on in-context learning (ICL) using hybrid architectures built from existing large language models, but its contributions are limited, lacking both novelty and clear motivation. While the open-source codebase is a positive aspect, the study's results are difficult to interpret, and the paper fails to offer meaningful insights or intuition behind performance variations. Additionally, the writing is unclear, with important details either missing or poorly explained.

As a summary, this paper is rejected due to the following reasons:
1. The paper offers little in terms of technical innovation or new methodologies.
2. The rationale for exploring these particular hybrid architectures is not well-articulated, diminishing the study's significance.
3. Clarity issues in both the presentation and explanation of results make it difficult to derive value from the findings.